# Short- and Long-Interval Prime-Boost Vaccination with the Candidate Vaccines MVA-SARS-2-ST and MVA-SARS-2-S Induces Comparable Humoral and Cell-Mediated Immunity in Mice

**DOI:** 10.3390/v15051180

**Published:** 2023-05-17

**Authors:** Georgia Kalodimou, Sylvia Jany, Astrid Freudenstein, Jan Hendrik Schwarz, Leonard Limpinsel, Cornelius Rohde, Alexandra Kupke, Stephan Becker, Asisa Volz, Alina Tscherne, Gerd Sutter

**Affiliations:** 1Division of Virology, Department of Veterinary Sciences, LMU Munich, 85764 Oberschleißheim, Germany; georgia.kalodimou@viro.vetmed.uni-muenchen.de (G.K.); sylvia.jany@viro.vetmed.uni-muenchen.de (S.J.); astrid.freudenstein@viro.vetmed.uni-muenchen.de (A.F.); alina.tscherne@viro.vetmed.uni-muenchen.de (A.T.); 2German Center for Infection Research (DZIF), Partner Site Munich, 85764 Oberschleißheim, Germany; 3Institute of Virology, Philipps University of Marburg, 35043 Marburg, Germany; rohdecor@staff.uni-marburg.de (C.R.); kupke@staff.uni-marburg.de (A.K.); becker@staff.uni-marburg.de (S.B.); 4German Center for Infection Research (DZIF), Partner Site Gießen-Marburg-Langen, 35043 Marburg, Germany; 5Institute of Virology, University of Veterinary Medicine Hannover, 30559 Hannover, Germany; asisa.volz@tiho-hannover.de; 6German Center for Infection Research (DZIF), Partner Site Hannover-Braunschweig, 30559 Hannover, Germany

**Keywords:** Modified Vaccinia virus Ankara, SARS-CoV-2, preclinical testing, vaccinia virus, vector vaccine

## Abstract

The COVID-19 pandemic caused significant human health and economic consequences. Due to the ability of SARS-CoV-2 to spread rapidly and to cause severe disease and mortality in certain population groups, vaccines are essential for controlling the pandemic in the future. Several licensed vaccines have shown improved protection against SARS-CoV-2 after extended-interval prime-boost immunizations in humans. Therefore, in this study, we aimed to compare the immunogenicity of our two Modified Vaccinia virus Ankara (MVA) based COVID-19 candidate vaccines MVA-SARS-2-S and MVA-SARS-2-ST after short- and long-interval prime-boost immunization schedules in mice. We immunized BALB/c mice using 21-day (short-interval) or 56-day (long-interval) prime-boost vaccination protocols and analyzed spike (S)-specific CD8 T cell immunity and humoral immunity. The two schedules induced robust CD8 T cell responses with no significant differences in their magnitude. Furthermore, both candidate vaccines induced comparable levels of total S, and S2-specific IgG binding antibodies. However, MVA-SARS-2-ST consistently elicited higher amounts of S1-, S receptor binding domain (RBD), and SARS-CoV-2 neutralizing antibodies in both vaccination protocols. Overall, we found very comparable immune responses following short- or long-interval immunization. Thus, our results suggest that the chosen time intervals may not be suitable to observe potential differences in antigen-specific immunity when testing different prime-boost intervals with our candidate vaccines in the mouse model. Despite this, our data clearly showed that MVA-SARS-2-ST induced superior humoral immune responses relative to MVA-SARS-2-S after both immunization schedules.

## 1. Introduction

The global pandemic triggered by severe acute respiratory syndrome coronavirus 2 (SARS-CoV-2) resulted in the rapid development, testing, and licensing of several vaccines [1,2,3,4,5]. The first COVID-19 vaccines were granted emergency use authorization within the first year of the SARS-CoV-2 pandemic. Due to the high demand and lack of supply, some governments chose to extend the time interval between the prime and booster immunization in order to increase the number of doses available for primary vaccination [6,7,8,9]. However, when these decisions were made, only limited data on the efficacy of an extended interval between prime and booster immunization of COVID-19 vaccines were available [10]. For instance, data from the phase 3 clinical trial for the ChAdOx1 nCoV-19 vaccine (AZD1222) showed better effectiveness when the booster was given at least 6 weeks after the prime immunization [2]. However, there was no information on the efficacy of the mRNA vaccines when using an extended time interval, as the phase 3 trials for these vaccines were only tested at 3–4-week intervals [1,4]. Subsequent population cohort studies performed during the COVID-19 vaccine rollout suggested that a longer prime-boost interval was in fact also beneficial for the efficacy of mRNA vaccines [11,12,13]. Considering these findings, it is of great interest to test the effects of extended prime-boost immunization protocols on other COVID-19 vaccines to determine if their efficacy against SARS-CoV-2 infection can be enhanced.

Modified Vaccinia virus Ankara (MVA) is a highly attenuated strain of vaccinia virus that was generated by more than 500 serial passages in chicken embryonic fibroblasts (CEF). Due to its strict attenuation, MVA lost the ability to replicate in most mammalian cells and as a consequence exhibits an excellent safety profile as a vaccine for use in humans [14,15,16,17,18]. Despite its replication deficiency, the synthesis of viral proteins is unaffected in MVA-infected mammalian cells, enabling them to produce large amounts of recombinant protein [14,16]. Previously, we used this platform to develop two COVID-19 candidate vaccines, targeting the full-length spike (S) protein of SARS-CoV-2 [19,20]. One vaccine, MVA-SARS-2-S, expressed the native form of the S protein [19], whereas the second vaccine, MVA-SARS-2-ST, expressed a modified prefusion stabilized form of the S protein [20]. Both vaccines were highly immunogenic when tested in a preclinical mouse model, inducing strong and robust CD8 T cell and antibody responses. In addition, MVA-SARS-2-ST induced more broadly reactive S-specific humoral immunity and demonstrated enhanced protection against SARS-CoV-2 compared to MVA-SARS-2-S in the Syrian hamster model and the lethal K18-hACE2 mouse model [20]. Hereby, MVA-SARS-2-ST mediated delivery of a stabilized prefusion S antigen that influences the enhanced S1-specific and neutralizing antibody responses, whereas the S1 dissociation from S2 in the native S protein produced from MVA-SARS-2-S reduces the quantity and quality of S1- and RBD- specific antibodies. 

In previous studies, we tested our COVID-19 candidate vaccines using our standardized 21-day prime-boost interval in mice and hamsters. For the current study, we compared the immunogenicity of our two candidate vaccines after the standard 21-day and the extended 56-day prime-boost immunization protocols using the recommended (full) human dose in BALB/c mice. Overall, the two immunization schedules induced comparable SARS-CoV-2 S-specific humoral and CD8 T cell-mediated immunity, suggesting that the chosen intervals may not be suitable for comparing short and extended prime-boost schedules with our candidate vaccines in the mouse model. Despite this, our results clearly demonstrated that MVA-SARS-2-ST induced superior humoral immunity relative to MVA-SARS-2-S after the 21-day and 56-day vaccination protocols. Our work provides a foundation for future studies testing the effects of prime-boost interval extensions on the efficacy of our COVID-19 candidate vaccines.

## 2. Materials and Methods

### 2.1. Plasmid Construction

The coding sequence of SARS-CoV-2 spike protein (SARS-2-S) (GenBank accession number MN908947.1) was modified in silico to remove guanine or cytosine runs and termination sequences for vaccinia virus (VACV) early transcription. For the native form of SARS-2-S, a C-terminal HA-tag sequence (YPYDVPDYA, aa 98–106 from influenza virus) [19,21] was added to the gene sequence. To generate a prefusion stabilized form of SARS-CoV-2 S (SARS-2-ST), the sequence was modified by introducing five mutations to stabilize the prefusion conformation of the spike protein as described previously [20] (Figure 1a). The modified SARS-2-S and SARS-2-ST cDNA were generated by DNA synthesis (Eurofins Genomics, Ebersberg, Germany) and were cloned into the MVA transfer plasmid pIIIH5red under control of the VACV early/late promoter PmH5 [22].

### 2.2. Generation of Recombinant Viruses

MVA vector viruses were generated using established protocols as described in previous studies [17,18,19,20,23]. Briefly, non-recombinant MVA (clonal isolate MVA-F6-sfMR) served as a backbone to generate MVA vector viruses expressing SARS-2-S or SARS-2-ST sequences. Monolayers of DF-1 or CEF at 90–95% confluence were infected with MVA-F6-sfMR at a multiplicity of infection (MOI) of 0.05 and transfected with the pIIIH5red-SARS-2-S or pIIIH5red-SARS-2-ST plasmid DNA using X-tremeGENE HP DNA Transfection Reagent (Roche Diagnostics, Penzberg, Germany) according to the manufacturer’s protocol. Recombinant viruses were identified by co-expression of the fluorescent marker mCherry and clonally isolated by serial plaque purification. To generate vaccine preparations, MVA-SARS-2-S and MVA-SARS-2-ST were amplified on CEF and DF-1 cell monolayers, purified by ultracentrifugation using a 36% sucrose cushion, and reconstituted to high-titer stocks in Tris-buffered saline (pH 7.4). Viral titers were determined by counting plaque-forming units (PFU) on CEF monolayers [23].

### 2.3. Vaccination Experiments in Mice

Female BALB/c mice aged 6–9 weeks were obtained from Charles River Laboratories (Sulzfeld, Germany) and kept in specific pathogen-free conditions. Animals had access to food and water ad libitum and were allowed to adapt to the facility for at least one week before starting the experiments. All animal experiments were performed in compliance with the European and national regulations for animal experimentation (European Directive 2010/63/EU; Animal Welfare Acts in Germany) and approval was obtained from regional animal ethics authorities. Mice were immunized with 10^8^ PFU of recombinant MVA-SARS-2-S, MVA-SARS-2-ST, non-recombinant MVA, or saline (PBS) in the quadriceps muscle of the left hind leg. Mice were immunized following either a 21-day (short-term/standard) or a 56-day (long-term/extended) prime-boost schedule (Figure 1b). For the short-term schedule, blood was collected on days 0, 18, and 35, and for the long-term schedule on days 0, 53, and 70. Serum was collected from coagulated blood that was centrifuged at 2000 rpm for 10 min in MiniCollect vials (Greiner Bio-One, Frickenhausen, Germany) and stored at −20 °C until use.

### 2.4. Antigen-Specific IgG Enzyme-Linked Immunosorbent Assay (ELISA)

SARS-CoV-2 spike-specific antibodies were measured by ELISA as described previously [19,20,24]. Briefly, flat bottom 96-well ELISA plates (Nunc MaxiSorp Plates, Thermo Fisher Scientific, Karlsruhe Germany) were coated with 50 ng/well of one of the SARS-CoV-2 spike antigens listed in Appendix A (from ACROBiosystems, Newark DE, USA and The Native Antigen Company, Kidlington, UK) overnight at 4 °C. Mouse sera were three-fold serially diluted in PBS containing 1% BSA (Carl Roth GmbH, Karlsruhe, Germany) (PBS/BSA), transferred to the coated ELISA plates and incubated for 1 h at 37 °C. Plates were then probed with goat anti-mouse IgG HRP diluted in PBS/BSA. SARS-CoV-2 spike-specific antibodies were detected using 3′3′,5′5′- Tetramethylbenzidine (TMB) Liquid Substrate System for ELISA (Sigma-Aldrich, Taufkirchen, Germany), and the reaction was stopped using Stop Reagent for TMB Substrate (Sigma-Aldrich). The absorbance was measured at 450 nm with a 620 nm reference wavelength using the Spark^®^ multimodal microplate reader (Tecan, Männedorf, Switzerland). Data were normalized using a positive control sample. The cut-off value for positive mouse serum samples was determined by calculating mean OD 450 nm values of the PBS control group sera plus six standard deviations (mean + 6 SD).

### 2.5. SARS-CoV-2 Virus Neutralization Test 100 (VNT_100_)

The neutralizing capacity of mouse sera was determined as described previously [19,20,25]. In summary, samples were serially diluted and incubated with 100 50% tissue culture infectious dose (TCID_50_) of SARS-CoV-2 (BavPat1/2020 isolate, European Virus Archive Global # 026V-03883). Four days after infection, Vero E6 cells (ATCC CRL-1586) were observed for cytopathic effects (CPE) by microscopy. Neutralization was defined as the absence of CPE compared to the virus controls.

### 2.6. Measurement of Cellular Response by Enzyme-Linked Immunospot (ELISPOT)

Immunized mice were sacrificed 14 days post-booster immunization and spleens were collected and prepared as published previously [19,26]. IFN-γ producing cells were measured by ELISPOT assay according to the manufacturer’s protocol (Mouse IFN-γ ELISpot PLUS kit (ALP), Mabtech, Nacka Strand, Sweden). Briefly, splenocytes were plated on 96-well plates and re-stimulated with the SARS-2 peptide S_268–276_ (GYLQPRTFL) [19] (final concentration 2 µg/mL) for 48 h. Non-treated cells and cells treated with phorbol myristate acetate (PMA) plus ionomycin (PMA/Ionomycin) (both from Sigma-Aldrich, Taufkirchen, Germany) or with 2 µg/mL of VACV peptide F2_26–34_ (SPYAAGYDL) [27] served as controls. Plates were stained as described in the manufacturer’s protocol and the spots were counted using an automated ELISPOT plate reader (ELi. Scan, A.EL.VIS, Hannover, Germany), using ELISPOT analysis software (Eli-Analyze, A.EL.VIS).

### 2.7. Intracellular Cytokine Staining (ICS)

Intracellular cytokine staining was performed as published previously [19,24]. Briefly, splenocytes were stimulated with SARS-2 S peptide S_268–276_ (GYLQPRTFL) [19] or VACV peptide F2_26–34_ (SPYAAGYDL) [27] to analyze SARS-2-S and MVA-specific CD8 T cells respectively. Non-treated cells and cells stimulated with PMA/ionomycin served as controls. Two hours after commencing the stimulation, Brefeldin A (Biolegend, San Diego, CA, USA) was added and cells were incubated for another 4 h. After stimulation, cells were stained extracellularly with anti-mouse CD3 phycoerythrin (PE)-Cy7, anti-mouse CD4 Brilliant Violet 421, anti-mouse CD8α Alexa Fluor 488 and purified CD16/CD32 (all from Biolegend, Appendix A). Cells were washed, fixed, permeabilized, and stained with anti-mouse IFN-γ allophycocyanin (APC) and anti-mouse TNF-α PE (both from Biolegend, Appendix A). Data were acquired using the MACSQuant VYB Flow Analyzer (Miltenyi Biotec, Bergisch Gladbach, Germany) or the NovoCyte Quanteon (Agilent Technologies, Waldbronn, Germany) and analyzed using FlowJo software (FlowJo LLC, Ashland, OR, USA).

### 2.8. Statistical Analysis

Data were prepared using GraphPad Prism version 5 (GraphPad Software Inc., Boston, MA, USA) and were expressed as the mean ± standard error of the mean (SEM) or geometric mean ± 95% confidence interval (CI). ELISA and virus neutralization data were log-transformed prior to statistical analysis. Data were analyzed by two-tailed *t*-tests for comparing two groups and one-way ANOVA to compare three or more groups. The threshold for statistical significance was *p* < 0.05.

## 3. Results

### 3.1. Humoral Immunity Induced by High Doses of MVA-S and MVA-ST after Short- and Long-Interval Prime-Boost Immunization Schedules

We previously generated and characterized the immunogenicity of two SARS-CoV-2 candidate vaccines, MVA-SARS-2-S (MVA-S) and MVA-SARS-2-ST (MVA-ST), expressing the native or prefusion stabilized forms of the SARS-CoV-2 spike (S) protein respectively [19,20]. In the current study, we tested the effects of short-term and long-term prime-boost schedules in the BALB/c preclinical mouse model, using the standard human dose of 10^8^ plaque-forming units (PFU) via the intramuscular route [19,20]. Sera from vaccinated female mice were analyzed for S antigen-specific IgG binding antibodies, 14 days after the second immunization by ELISA (Figure 2). MVA-S and MVA-ST induced high full-length S- and S2-specific IgG binding antibody titers after both immunization schedules (Figure 2a,d). Moreover, MVA-ST generated significantly higher S1- and S receptor binding domain (RBD)-specific IgG binding titers than MVA-S after the 21-day and 56-day vaccination protocols (Figure 2b,c), supporting previous observations [20]. When we compared the effects of short and extended prime-boost intervals on the serum IgG response, we did not find significant differences in full-length S, S1, and S2-specific IgG binding titers after the two different immunization protocols for each vaccine. Interestingly, a significant difference in serum S RBD-specific IgG binding titers was observed in the MVA-S group, in which the geometric mean of the binding titers reduced from 1:785 after the 21-day schedule to 1:208 after the 56-day schedule. In the MVA-ST group, however, we noticed a small but non-significant reduction in binding antibodies. The geometric mean of the S RBD IgG binding titers was 1:27,877 after the 21-day schedule and 1:20,234 after the 56-day schedule. When we assessed the serum IgG response after a single immunization with MVA-S and MVA-ST, we found that the serum IgG binding titers against full-length S, S1, and S RBD were overall significantly higher at day 53 post-prime immunization (56-day schedule) than at day 18 post-prime-immunization (21-day schedule) (Appendix A). This indicated that one dose of our candidate vaccines stimulated an antibody response that was still detectable 8 weeks after prime immunization.

Next, we measured SARS-CoV-2-specific neutralizing antibody titers by VNT_100_ assay. When we compared the two immunization schedules for each candidate vaccine, no statistically significant differences were observed (Figure 2e). However, when comparing the two candidate vaccines to each other after the short and extended prime-boost schedules, we discerned significant differences. MVA-ST stimulated a significantly stronger SARS-CoV-2 neutralizing antibody response compared to MVA-S. The geometric mean of the neutralizing titers after the 21-day and 56-day schedules were 1:734 and 1:451 for MVA-ST respectively, and 1:16 and 1:22 for MVA-S respectively. 

Taken together, these results indicate that extending the prime-boost schedule from 21 days to 56 days did not have a significant effect on the humoral immune response generated by our candidate vaccines. Moreover, our results also showed that after both immunization schedules, MVA-ST stimulated stronger SARS-CoV-2 spike-specific humoral immunity than the MVA-S vaccine.

### 3.2. CD8 T Cell-Mediated Immunity Induced by MVA-S and MVA-ST after Short- and Long-Interval Prime-Boost Immunization Schedules

After studying the humoral immune response induced by our two immunization schedules, we next analyzed their effects on S-specific T cell-mediated immunity. Splenocytes obtained from the mice used in the above humoral immunity study were stimulated with the S peptide S_268–276_ (GYLQPRTFL) and analyzed for cytokine secretion by IFN-γ ELISPOT and ICS assays. Mixed responses were observed after the 21-day and 56-day prime-boost immunization schedule (Figure 3). After both immunization schedules, the mean spot-forming colony (SFC) counts were slightly higher in the MVA-ST group relative to the MVA-S group, although the differences were not statistically significant (Figure 3a,d). The SFC counts were 1177 and 1227 SFC/10^6^ cells for MVA-ST after 21-day and 56-day prime-boost immunization protocols respectively, and 897 and 1007 SFC/10^6^ cells for MVA-S after the 21-day and 56-day schedules respectively. A similar non-statistically significant trend was also demonstrated by ICS after the 21-day but not the 56-day vaccination schedule (Figure 3b,e). Interestingly, the absolute numbers of IFN-γ^+^ CD8 T cells were lower after the 56-day schedule relative to the 21-day schedule, which contrasted with the ELISPOT results. When we analyzed the cytokine profiles of the S-specific CD8 T cells, we found that the majority of IFN-γ-producing CD8 T cells induced by MVA-ST and MVA-S also produced TNF-α after both immunization schedules (Figure 3c, f and Appendix A). In fact, over 75% of cytokine-producing CD8 T cells were IFN-γ^+^TNF-α^+^ in the MVA-ST and MVA-S groups. When we analyzed MVA-specific T-cell immunity after our 21-day and 56-day prime-boost immunization schedules, we observed identical trends (Appendix A).

## 4. Discussion

Our study compared the humoral and cellular immune responses of two candidate vaccines, MVA-ST, and MVA-S, after 21- and 56-day prime-boost intervals. We showed that both immunization schedules overall induced comparable humoral immune responses when we immunized female BALB/c mice with a dose of 10^8^ PFU of the experimental vaccines. The effects of extending the time interval between priming and boosting have been previously studied using viral vector vaccines, protein vaccines, and more recently mRNA vaccines. These studies have yielded different results, which are related to the type of vaccine or the prime-boost protocols used.

In the context of poxvirus vector vaccines, several studies have demonstrated the benefits of extended prime-boost schedules on vaccine-mediated immunity. A phase 1/2a clinical trial that tested the immunogenicity of an MVA viral vector vaccine targeting the H5 influenza A virus subtype (MVA-H5-sfMR) showed that extending the interval between prime and booster immunization enhanced the influenza neutralizing antibody response [18]. All volunteers who were given a booster shot 1 year after the prime immunization generated strong anti-influenza humoral immune responses. In fact, the sera of volunteers who received one immunization with MVA-H5-sfMR followed by a booster 1 year later displayed elevated hemagglutinin inhibition against homologous and heterologous influenza strains relative to sera from volunteers who received two doses of the vaccine (4 weeks apart) followed by a booster 1 year later [18]. Similarly, in an HIV vaccine phase 3 trial testing heterologous immunization using ALVAC-HIV (vCP1521) (a canarypox vector vaccine) and AIDSVAX^®^ B/E (a bivalent protein vaccine) showed enhanced humoral immunity when the booster was given at a later time point [28]. When ALVAC-HIV (vCP1521) and AIDSVAX^®^ B/E were given together 15 or 18 months after the primary immunizations, the volunteers demonstrated improved virus-neutralizing abilities relative to those who received the booster 12 months after the primary immunizations [28]. Finally, in a preclinical non-human primate (NHP) study, in which MVA-specific immunity was tested using 14-day and 58-day prime-boost immunization schedules, the extended schedule was associated with significantly higher MVA-specific serum IgG levels and enhanced MVA neutralization [29]. In fact, a clinical trial testing compressed MVA prime-boost regimens found that intervals of less than 21 days impaired the humoral immune response [30]. By contrast, our 21-day and 56-day prime-boost vaccination schedules did not significantly affect the overall strength of the humoral immune response against SARS-CoV-2, nor did it result in an overt increase in SARS-CoV-2 neutralization. This discrepancy may be related to the different time schedules that we used for our vaccinations. It is possible that the extension of our prime-boost vaccination protocol from 21 days to 56 days was not sufficient to observe any effects in our mouse model.

Using a protein-based vaccine against the H56 tuberculosis antigen, Pettini et al. [31] found that extending the time interval between prime and boost from 4 weeks to 18 weeks in mice did not affect antigen-specific serum IgG levels. Despite this, the extended interval vaccination protocol induced a higher number of germinal center (GC) B cells and antigen-specific plasma cells [31]. Kinetics analysis revealed that 4 weeks after the first vaccination the GC B cell response was still in the effector phase, whereas by 18 weeks it had declined [31]. These findings have been supported by a recent study analyzing the B cell immunity after prime-boost immunization with COVID-19 mRNA vaccines in humans. Extending the time between prime and boost from the manufacturer’s recommended 3–4-week interval to 16 weeks improved the magnitude and the maturity of the S-specific B cell response [32]. This suggests that different intervals between immunizations can modulate the B cell response in the absence of overt changes to the serum IgG response. It should be noted that the type of vaccine used affects the kinetics of the GC B cell response [33,34,35]. For example, evidence suggests that the peak GC response after immunization with a recombinant adenovirus vaccine peaks at around 7 days but is still detectable after 37 days [34]. Thus, in relation to our candidate vaccines MVA-S and MVA-ST, a booster at day 21 may coincide with the contraction phase of the GC B cell response and the 56-day booster is likely to occur after the response has declined.

In the context of prime-boost dosing intervals and licensed COVID-19 vaccines, a recent multi-center prospective cohort study assessing hospital workers immunized with the BNT162b2 mRNA vaccine using short (3–4 week) and extended (6–14 week) vaccination schedules found a time-interval dependent effect on humoral immunity in vaccinated, SARS-CoV-2 naïve volunteers [11]. Overall, the SARS-CoV-2 naïve extended interval group showed significantly elevated SARS-CoV-2 neutralization of the wild-type virus and the alpha, beta, gamma, and delta variants, enhanced ACE2 inhibition and significantly higher spike and spike RBD IgG binding titers [11]. Interestingly, this advantage was not observed in volunteers who had been infected with SARS-CoV-2 prior to vaccination. In addition, the authors found that significant differences in anti-S antibody responses were only observed after larger dosing intervals. For example, comparisons between 4-week and 10–12-week immunization intervals in the SARS-CoV-2 naïve volunteers yielded significant differences, whereas shorter interval differences, such as 6–8 weeks versus 10–12 weeks, did not show any effect [11].

The benefits of extended dosing intervals have also been demonstrated with the COVID-19 vaccine ChAdOx1 nCoV-19 (AZD1222). A pooled analysis of randomized clinical trials with this vaccine demonstrated that a prime-boost vaccination interval of 12 weeks or more induced elevated SARS-CoV-2 binding and neutralizing antibodies and afforded better protection against the development of symptomatic infections relative to intervals of less than 6 weeks [36]. Thus, it would be interesting to determine if comparable trends are also observed after larger dosing intervals with our highly immunogenic candidate vaccine, MVA-ST.

Like vaccine-induced humoral immunity, evidence suggests that increasing the interval between priming and boosting enhances vaccine-mediated T-cell immunity. Using an alpha replicon-based experimental vaccine, Knudsen et al. [37] found that increasing the interval between priming and boosting from 1- to 3-, 6- or 9-weeks significantly enhanced the magnitude of the antigen-specific CD8 T cell response in mice at 5 weeks post-boost. Moreover, comparisons between the 1- and 9-week schedules showed that the latter induced more CD62L-CD127+ effector memory T cells and CD27+CD43- memory T cells [37]. Extrapolating from this study to ours, however, is difficult because of the dosing interval tested. At 1-week post-prime immunization, the T cell response is in the expansion phase, whereas by 3 weeks, the time of our short-term booster vaccination, it is in the contraction phase [38]. Another study, which tested the immunogenicity of an experimental heterologous adenovirus 5 (Ad5) and VACV-based malaria vaccine, demonstrated that increasing the prime-boost interval from 2 weeks (the peak of T cell expansion) to 8 weeks also enhanced CD8 T cell immunity and provided greater protection against challenge [39]. The 8-week prime-boost immunization protocol also generated more activated antigen-specific CD8 T cells and fewer apoptotic T cells than the 2-week protocol [39]. This contrasted with our study, in which the longer immunization protocol did not significantly increase the magnitude of the S-specific CD8 T cell response nor did it enhance their polyfunctionality. The lack of overt changes to the magnitude of S-specific CD4 and CD8 T cell responses after extended prime-boost immunization protocols has also been observed in humans vaccinated with COVID-19 mRNA vaccines [32]. Unfortunately, we could not detect S-specific CD4 T cells in our study and thus we could not determine if an identical trend occurred in this population after the long-term prime-boost immunization protocol.

Taken together, our study showed that extending the interval between priming and boosting from 21 to 56 days did not have a significant effect on the SARS-CoV-2 S-specific humoral and CD8 T cell response. Despite this, our data clearly showed that MVA-ST induced superior SARS-CoV-2 S-specific humoral immunity relative to MVA-S after both vaccination protocols. In line with our previous studies, the stabilized prefusion S antigen delivered by MVA-ST allows for the induction of enhanced S1-specific and neutralizing antibody responses [20]. In contrast, the authentic proteolytic processing of the native S protein produced by MVA-S causes S1 dissociation from S2 and reduces the quantity of S1- and RBD-specific antibodies.

In the future, it would be of interest to test longer-extended prime-boost vaccination protocols to determine if the non-significant trends we observed are further enhanced. It would also be beneficial to further characterize short-and long-term schedules in other small animal models such as hamsters. Finally, it would be valuable to further characterize vaccine immunogenicity by analyzing the B and T cell responses in more detail to determine if there are qualitative differences in immune responses in the absence of overt changes to their magnitude. It is important to consider the time interval between immunizations during preclinical testing of our COVID-19 candidate vaccines, as it will enable us to develop new and improved protocols to maximize their efficacy against circulating and new emerging variants.

## Figures and Tables

**Figure 1 viruses-15-01180-f001:**
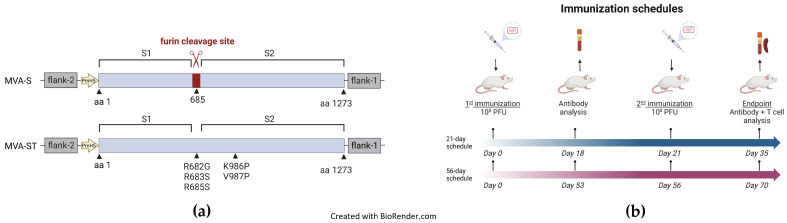
Schematic diagrams of the MVA-SARS-2-ST and MVA-SARS-2 S vaccines and immunization schedules. (**a**) Schematic representation of MVA-SARS-2-S (MVA-S) and MVA-SARS-2-ST (MVA-ST). MVA-S encodes for the native spike sequence containing the furin cleavage site [19]. MVA-ST encodes for the spike protein with the five indicated mutations for prefusion stabilization [20]. Recombinant spike sequences were cloned into deletion site III of non-recombinant MVA under transcriptional control of the PmH5 promoter. (**b**) Schematic diagram of the 21-day (short-term) and the 56-day (long-term) immunization schedules. Mice were immunized twice with 10^8^ PFU via the intramuscular route. Blood was collected for antibody analysis at the indicated time points. At the endpoint, spleens were collected for T-cell analysis.

**Figure 2 viruses-15-01180-f002:**
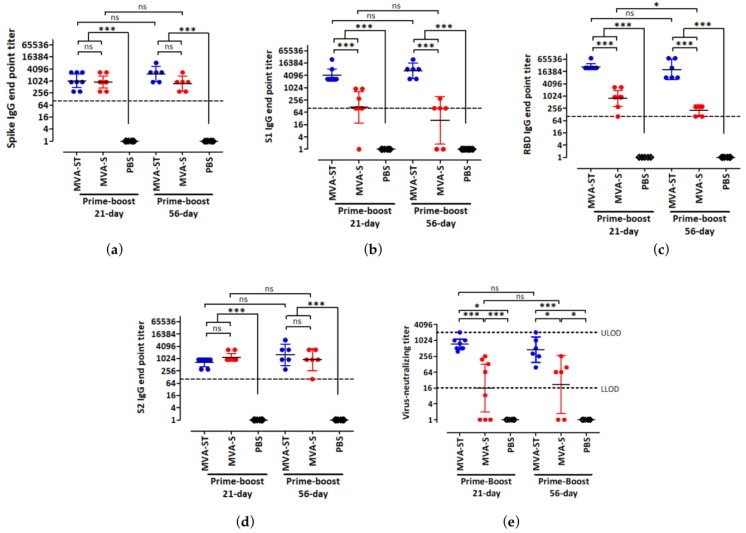
Spike antigen-specific antibodies induced by two vaccination protocols with MVA-SARS-2-ST and MVA-SARS-2-S. Groups of BALB/c mice (n = 6 to 8) were immunized twice over a 21-day interval or a 56-day interval with 10^8^ PFU of MVA-SARS-2-ST (MVA-ST) or MVA-SARS-2-S (MVA-S) via the intramuscular (i.m.) route. Mice inoculated with saline (PBS) served as controls. Sera were collected 14 days after the booster immunization and analyzed for SARS-2 full-length spike (**a**), S1 (**b**), receptor binding domain (RBD) (**c**), and S2 (**d**) specific IgG binding titers by ELISA. Graphs a–d show normalized data after prime-boost immunization and dashed lines represent the limits of detection (LOD). (**e**) VNT_100_ was determined after prime-boost immunization. Dashed lines represent lower (L)LOD and upper (U)LOD. Bars represent the geometric mean + 95% confidence interval (CI). For statistical analysis, log-transformed data were analyzed by one-way ANOVA and Tukey posttest. Asterisks represent statistically significant differences between the two groups. *** *p* < 0.001, * *p* < 0.05, ns = not significant.

**Figure 3 viruses-15-01180-f003:**
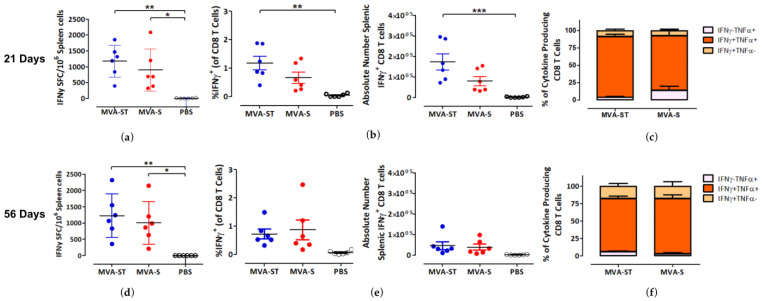
Activation of SARS-2-S specific CD8 T cells after prime-boost immunization (21-day and 56-day intervals) with MVA-SARS-2-ST and MVA-SARS-2-S. Groups of BALB/c mice (n = 6) were vaccinated with 10^8^ PFU of MVA-SARS-2-ST (MVA-ST) and MVA-SARS-2-S (MVA-S) over a 21-day interval (**a**–**c**) or a 56-day interval (**d**–**f**) via the i.m. route. Saline-immunized mice (PBS) served as a control. Splenocytes were collected and prepared on day 14 after the booster immunization. Total splenocytes were stimulated with the H2-d restricted peptide of the SARS-2-S protein S_268–276_ (GYLQPRTFL) and were measured by IFN-γ ELISPOT assay and IFN-γ and TNF-α intracellular cytokine staining (ICS) plus FACS analysis. (**a**,**d**) IFN-γ spot forming colonies (SFC) for stimulated splenocytes measured by ELISPOT assay. (**b**,**e**) IFN-γ production by CD8 T cells measured by FACS analysis. Graphs show the frequency and absolute number of IFN-γ+ CD8 T cells. (**c,f**) Cytokine profile of S_268–276_-specific CD8 T cells. Graphs show the mean frequency of IFN-γ^−^TNF-α^+^, IFN-γ^+^TNF-α^+^, and IFN-γ^+^TNF-α^−^ cells within the cytokine-positive CD8 T cell compartment. Bars represent the mean + standard error of the mean (SEM). Differences between groups were analyzed by one-way ANOVA and Tukey posttest. Asterisks represent statistically significant differences between the two groups. *** *p* < 0.001, ** *p* < 0.01, * *p* < 0.05.

## Data Availability

The data that support the findings of this study are available from the corresponding author upon reasonable request.

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
