# Peer review of "Short- and Long-Interval Prime-Boost Vaccination with the Candidate Vaccines MVA-SARS-2-ST and MVA-SARS-2-S Induces Comparable Humoral and Cell-Mediated Immunity in Mice"

_viruses, 2023, doi:10.3390/v15051180_

Round 1
Reviewer 1 Report
Viruses-2401380
The manuscript by Georgia Kalodimou, et al., “Short and long interval prime-boost vaccination with the candidate vaccines MVA-SARS-2-ST and MVA-SARS-2-S induce comparable humoral and cell-mediated immunity in mice” examines and compares 2 previously developed MVA-Based SAR-CoV-2 “COVID-19” vaccine candidates (MVA-SARS-2-S (native spike S encoding) and MVA-SARS-2-ST (stabilized spike S encoding) using a short (21d) and long-interval (56d) prime-boost immunization schedule in mice. The authors analyze both IgG responses to Spike, RBD, S1 and S2, neutralizing titers; and also spike specific CD8 T cell responses.
The paper is very well-written and referenced, and data/figures clear. The overall conclusion is that the 21 vs 56-day (extended) prime to boost interval in the mouse, at least, is highly comparable in terms of the immune response and this (mouse) model is not suitable for analyzing or improving a vaccine regiment by modifying the intervals of a prime-boost strategy. The data also confirm and extend on the observation that a stabilized S spike immunogen is superior in inducing RBD, and virus-neutralizing sera titers.
Specific Comments:
1. The abstract states: “The short- and long-interval schedules induced robust but comparable S-specific CD8 T cell responses”. This is clear, but doesn’t the MVA-SARS-2-ST vaccine also elicit significantly better T cell responses in comparison to MVA-SARS-2-S? (Figure 3)
2. Also confusing, the abstract states: “both candidate vaccines induced similar levels of total S, S1-, S receptor binding domain (RBD)- and S2-specific IgG binding antibodies and SARS-CoV-2 neutralizing antibodies.” But Figure 2 indicates only the total anti-S spike and total anti-S2 levels are similar and not statistically different with either 21 or 56 day schedule; however the levels of anti-S1 and anti-RBD and virus-neutralizing titers are quite higher and statically significantly better in the MVA-SARS-2-ST immunized subjects as compared to MVA-SARS-2-S (Fig2-b,c,e). Why is this not drawn out in the abstract? Or discussion? These points seem relevant.
3. The authors might consider commenting on how/why there are higher titers to S1, RBD and neutralizing activity in the same subjects when compared to total S spike titer? Is this because of just steric hinderance or antibody competition to the native S spike as compared to individual RBD or S1 specific antibody binding?
4. Page 2, bottom, line 94, should be “as described previously (20)”
5. Figure 3, recommend adding labels to each row of figures: 21-day and 56-day on the left figure margin.
6. Page 8, line 335, should be “adenovirus vaccine peaks”
7. Page 9, line 356, should be “weeks or more”
8. Page 9, lines 361-362, perhaps merge first sentence with lower paragraph
Reviewer 2 Report
To assist the reader in navigating through your results, explain why there are no differences between the spike IgG or S2 IgG levels induced by MVA-ST and MVA-S whereas differences are seen for S1 IgG and RBD IgG.
The authors have stated that when MVA-specific T cell immunity was measured after 21- and 56-day prime-boost immunisation schedules, identical trends were observed as depicted in Supplementary Figures S2 and S3. However, it was not possible to understand what the Supplementary data was depicting without any figure legends and the labels in figures were not helpful either. It is essential that the authors include Figure legends for all the Supplementary figures. The quality of the supplementary figure presentation can be improved and made clearer.
It is not clear why the authors have suggested that the mouse model may not be suitable for testing MVA-SARS-2-ST and MVA-SARS-2-S on effects of different prime-boost intervals as they have done in this study. They need to provide a rationale to help the reader understand this line of reasoning. Perhaps if they waited for a longer period after the booster vaccine, say 100 days, would they have seen a difference? Some discussion will be necessary. Their previous study published in J Clin Invest. has clearly shown that responses generated by MVA-ST and MVA-S in mice and hamsters were comparable in that MVA-ST generated superior immune responses in both species.
It is not clear whether the BALB/c mice used in the study were males or females. Sex may have an influence on the immune response generated. This information should be included for the benefit of the reader.
Introduction, paragraph 2. The following sentence could be written more clearly. “Due to its strict attenuation, MVA lost the ability to replicate in most mammalian cells and thus, showing an excellent safety profile as a vaccine for human use …”. Perhaps they meant “Due to its strict attenuation, MVA lost the ability to replicate in most mammalian cells and as a consequence exhibits an excellent safety profile as a vaccine for use in humans …”.
